# Home Range and Habitat Use of Breeding Black-necked Cranes

**DOI:** 10.3390/ani10111975

**Published:** 2020-10-28

**Authors:** Wei Liu, Yuyi Jin, Yongjie Wu, Chenhao Zhao, Xingcheng He, Bin Wang, Jianghong Ran

**Affiliations:** 1Sichuan Key Laboratory of Conservation Biology on Endangered Wildlife, College of Life Sciences, Sichuan University, Chengdu 610064, China; 2019117@htu.edu.cn (W.L.); anastasia0728@hotmail.com (Y.J.); wuyongjie@scu.edu.cn (Y.W.); georgeharrisonmbe@gmail.com (C.Z.); hexingcheng@outlook.com (X.H.); 2College of Life Sciences, Henan Normal University, Xinxiang 453007, China; 3Institute of Ecology, China West Normal University, Nanchong 637002, China; wangbin513@outlook.com

**Keywords:** home range, habitat utilization, grazing activity, *Grus nigricollis*, marsh

## Abstract

**Simple Summary:**

The Black-necked Crane is the only crane that breeds on the Qinghai–Tibetan Plateau, and it is currently classified as a vulnerable species. Habitat destruction and loss are the main threats for the Black-necked Crane. However, the breeding needs of Black-necked Cranes with regard to their preferred habitat remain unknown. To find the driving factors of their habitat selection, we studied the utilization rate of three habitats within the home range of Black-necked Cranes during four consecutive breeding stages. Black-necked Cranes mainly utilize meadows in the whole breeding season, followed by marsh meadows and marshes. Compared with other stages, the utilization of marsh habitat slightly increased in the postfledging stage. Since this stage is a very vulnerable period for young cranes, we suggest that grazing needs to be managed during this stage, that is, from late May to early September.

**Abstract:**

To effectively protect a species, understanding its habitat needs and threats across its life-history stages is necessary. The Black-necked Crane (*Grus nigricollis*) is an endangered wetland bird species of the Qinghai–Tibetan Plateau, which is an important grazing area in China. To overcome the conflict between increasing grazing activities and the protection of wild cranes, we investigated the variation of habitat utilization within the home range of cranes at different stages (preincubation, incubation, postfledging, and fully fledged stages). We manually tracked 13 pairs of cranes in the Zoige international wetland, used the fixed-kernel-polygon (FKP) method to determine home-range size, and used satellite images to identify different habitat types. The average home-range size was 143.38 ± 34.46 ha. Cranes were most often located in meadow habitats followed by marsh meadows and marshes. During the postfledging stage, home-range size was significantly decreased, with the proportion of marsh habitat slightly increased. Since this stage is crucial for young-crane survival, research on the importance of marshes and effective protection measures should be further strengthened.

## 1. Introduction

From 1940 until today, the global number of breeding birds in grasslands has significantly declined [1,2]. Agricultural, and especially grazing, activities are mainly responsible for this decreasing trend [3]. The Qinghai–Tibetan Plateau in China is a biodiversity hotspot and one of the major Chinese pastoral areas. In recent years, an increase in grazing activities with a reduced area of wetlands has negatively affected the living and breeding environment of resident birds [4]. Finding a balance between grazing and the protection of rare species has been a focus of wildlife conservation on the Qinghai–Tibetan Plateau [5].

The home ranges of breeding birds in wetlands are typically composed of three habitats, marshes, marsh meadows, and meadows [6]. These habitats provide places for birds to mate, hatch [7], and forage [8]. Different habitats exert different effects on bird reproduction. For example, many waterfowl species typically nest and hatch in marshes, but forage in meadows that are some distance apart. The reason for this split of areas is the lower availability of food in marshes; therefore, these birds often forage in meadows that are farther away [9]. The composition of different habitats is very important for bird reproduction [10]. Breeding waterfowl are not randomly distributed, and marshes may be an important factor that affects their distribution [11]. Understanding the composition and size of different habitats in the home range, and their changes during different breeding stages, can help to clarify the distribution pattern of birds and the habitat requirements of breeding areas [12]. This knowledge helps the government to overcome the conflict between animal protection and grazing activities. For example, a win–win model can be achieved by adjusting grazing density and turn-out date [13].

The Black-necked Crane (*Grus nigricollis*) is an endangered wetland species on the Qinghai–Tibetan Plateau. The International Union for Conservation of Nature (IUCN) classified the Black-necked Crane as “vulnerable” due to its decreasing global population of currently 10,070–10,970 individuals [14]. Due to the plateau’s environmental inaccessibility for comprehensive and persistent field research, the Black-necked Crane remains the least-studied crane species in the world [15]. In the 1980s and 1990s, studies on the Black-necked Crane focused on the population characteristics in their wintering area, including preliminary ecological studies [16,17]. Up to the 21st century, research on the individual behavior, foraging habitat, breeding behavior, migration route, and genome of the Black-necked Crane has emerged [18,19,20,21].

In the Zoige wetland on the Qinghai–Tibetan Plateau, which is the most concentrated breeding area of Black-necked Cranes [22], the cranes have a relatively fixed territory during their breeding period [17]. They usually choose wetlands such as marshes to nest and hatch because these habitats are comparatively safer [23]. They often choose meadow habitats in the Zoige wetland for foraging [24] because these habitats can provide abundant invertebrate [24] and potentially vertebrate food resources [25]. However, habitat patches within the home range of Black-necked Cranes have not been investigated to date. The breeding needs of Black-necked Cranes with regard to their preferred habitat remain unknown, which is not conducive to their protection. Knowing their foraging area and breeding-area changes is the precondition to further study this species and better understand its interaction with the surrounding environment [26].

From March until November 2015, this study was conducted in the Zoige wetland, which is the most concentrated breeding location for Black-necked Cranes. The home range and inner habitat patches of the Black-necked Crane were investigated, with a particular focus on marsh-habitat demand during breeding season. The aims of this study were to (1) describe home-range size and changes among the four breeding stages (preincubation, incubation, postfledging, and fully fledged), (2) assess the variation of habitats within the home range during the four breeding stages, and (3) provide scientific support for the effective protection of Black-necked Cranes, and for the management of yak and Tibetan sheep husbandry in the Qinghai–Tibetan Plateau.

## 2. Materials and Methods

### 2.1. Study Area

The study was undertaken in the Zoige wetland (32°20′–34°05′ N, 101°36′–103°55′ E), which is located in the northeastern portion of the Qinghai–Tibetan Plateau (Figure 1). The Zoige wetland is one of the five most important grazing regions in China and a highland biodiversity hotspot inhabited by a number of endemic and endangered species [27]. The Zoige wetland is 7080 km^2^ in area, and the average elevation is 3400–3500 m above sea level. The climate of the Zoige wetland is typically influenced by mild-to-cold temperate continental monsoons. Annual mean temperature is approximately 0.7–1.1 °C, the highest monthly mean is 10.8 °C in July, and the lowest is −10.6 °C in January. Annual mean precipitation is 656.8 mm, 86% of which occurs between April and October. Total sunshine time is 2400 h per year, and total irradiance is about 580 kJ/cm^2^. Mean daily relative humidity is stable throughout the year at approximately 65%. The growing season of the marsh is very short, usually 5 months from May to September [28]. The Zoige wetland nature reserve was established in 1998 and covers an area of 1666 km^2^ for conserving the breeding habitats of water birds, especially Black-necked Cranes, and the vulnerable high-plateau wetland ecosystem [29]. The Zoige nature reserve has three types of function zones, the core (which is strictly protected), buffer (in which monitoring and research can be conducted), and experiment (in which other sustainable activities such as natural education and ecotourism can be conducted) zones (Figure 1c). Our research was undertaken in the two main breeding locations in the Zoige wetland: (1) the Flower marsh (33°53′–33°57′ N, 102°47′–102°54′ E, approximately 60 km^2^, Figure 1a) and (2) the Naleqiao marsh (33°29′–33°32′ N, 102°38′–102°42′ E, approximately 36 km^2^, Figure 1b) [30]. The habitats within our study area were classified as marsh, marsh meadow, and meadow [31]. We designed 4 sample plots in each habitat and ten sample quadrats in each sample plot (total sample quadrats = 360) to survey plant-community types, vegetation height, and water depth. The main plant-community type in the marsh habitats is *Carex muliensis*, vegetation height is 40.7 ± 2.5 cm, and water depth is 35.3 ± 0.9 cm. The main plant-community type in the marsh-meadow habitats is *Carex muliensis*–*Caltha scaposa*, vegetation height is 10.2 ± 0.5 cm, and water depth is 7.2 ± 0.5 cm. The main plant-community type in meadow habitats is *Kobresia tibetica*–*Kobresia capillifolia* and *Kobresia capillifolia*–*Potentilla anserina*, vegetation height is 5.6 ± 0.3 cm, and water depth is 0.8 ± 0.2 cm (see Appendix A).

### 2.2. Study Population and Identification of Breeding Families

We studied breeding cranes in four consecutive breeding stages from March until November 2015. The four breeding stages are preincubation (before the egg-laying date, March–June), incubation (from onset of egg laying to hatching date, April–July), postfledging (from the date of the nestling cranes leaving the nest to when nestling cranes start to fly (since cranes are precocial birds, chicks can leave the nest as soon as they are born); May–September), and fully fledged (from when nestling cranes fly until they depart on their migrations, July–October).

Prior to preincubation, we identified 23 pairs of cranes (14 pairs in the Flower marsh and 9 pairs in the Naleqiao marsh) within the study area. We used feather patterns (Figure 2) and location (cranes exhibiting preferences for specific sites when they arrive) to identify these pairs [32]. Not all pairs laid eggs or successfully hatched offspring; thus, we only collected data from 13 pairs at the fully fledged stage.

### 2.3. Locating Families and Data Collection

Underneath a camouflage coat, we manually tracked each breeding pair using a spotting scope (monocular telescope: Leica APO-TELEVID 82) at a minimal distance of 100–150 m from the cranes [33]. We recorded our position using a handheld global-positioning-system (GPS) receiver (Garmin GPSMAP 621sc, accuracy of ±5 m) and adjusted to the cranes’ location using laser range finders. We marked potential land markers (recognizable fixed objects such as artificial fences and tents) on a 1:50,000 local map (provided by the Ruoergai Forestry Bureau) to help confirm the direction. The method was consistent with animal-welfare standards described by Fair, et al. [34].

For each pair, we tracked each breeding stage four times except for when pairs had not laid eggs or when the offspring had died after hatching. Each tracking session lasted for 2 h, and crane positions were recorded every 5 min; thus, 24 data points were recorded for every 2 h tracking session. A new pair was selected for tracking after 2 h. Black-necked Cranes are diurnal, so all observations were conducted at 0800–1700 China Standard Time in 2015 [35]. We alternated experiment days between the Flower and Naleqiao marshes.

### 2.4. Home-Range Size, Overlap, and Site Fidelity

To calculate the area of different habitats in the home range, we first measured the home-range sizes of the breeding Black-necked Cranes. All consecutive breeding stages were studied to accurately map and understand their home-range requirements. We calculated home-range size by using position data in the home-range extension in the ArcView geographic-information system (GIS) 3.3 [36]. Home-range size was determined using the fixed-kernel-polygon (FKP) method with least-squares cross-validation [37] because the area within the boundary of an individual’s home range is generally disproportionately used [38]. We calculated the FKP home-range size with probability densities of 95% and 50% for each breeding pair to construct an accurate map of their home range and utilization distribution. An FKP of 50% represents the core area and is defined as the most highly utilized part of a home range, and 95% represents the whole home-range area [37]. ArcGIS (Environmental Systems Research Institute, Inc. 2005) was used to map the FKP-estimated home range of each pair of cranes within the study area.

We used an identity-linked generalized linear mixed model (GLMM) [39] to analyze the variation of home-range sizes between the four breeding stages; we specified pairs as the random effect, home-range size as the dependent variable, and breeding stage as the explanatory variable.

To understand the ecological needs of cranes for wetland resources, we calculated the home-range overlap (i.e., degree of overlap) between and within crane pairs (i.e., territoriality). First, we mapped the home-range distribution using ArcMap. Then, we estimated an overlap index to distinguish overlap differences between neighboring pairs of cranes. The degree of overlap was calculated using the following index [40]:$$\sqrt{\left(Area/Homerange1)\;\times \;((Area/Homerange2\right)},$$
where *Area* corresponds to the area of overlap between home-range pairs, and *Homerange1* and *Homerange2* correspond to the home range of subsequently named Families 1 and 2, respectively (i.e., pairs + offspring). A degree of overlap >50% was considered high [40]. We calculated the minimal percentage of overlap between two intersecting neighboring families on the basis of 50% and 95% FKP home-range estimators.

To assess site fidelity, we calculated the degree of overlap within the home range of 13 crane families throughout the four consecutive breeding stages (as per Doucette, 2010). We randomly selected one of the four breeding-stage home ranges as the numerator, and the home range of another stage as the base for calculating the level of site fidelity. In total, we calculated site-fidelity data for 6 × 13 groups in the four breeding stages for 13 families. The level of site fidelity was calculated using the following index [40]:$$\sqrt{\left(Area/Homerange1)\;\times \;(Area/Homerange2\right)},$$
where *Area* corresponds to the area of overlap within pairs of home ranges, and *Homerange1* and *Homerange2* correspond to the home ranges of two of the four breeding stages (i.e., preincubation and incubation, respectively). A degree of overlap >50% indicates a strong level of site fidelity [40].

### 2.5. Size and Proportions of Habitat Types within Crane Home Ranges

To investigate the size and proportions of habitat types within the home ranges of breeding pairs, we downloaded a satellite image (LC81310372015274BJC00, July 2015) from the Geospatial Data Cloud (http://www.gscloud.cn/sources). The image had a resolution of 30 × 30 m and covered the entire study area. The type of satellite imagery was Landsat 8 thematic mapper. We used the 2015 image because the field study was conducted in 2015. We classified the habitats into four categories: lake, marsh, marsh meadow, and meadow [31]. Vegetation interpretation was performed in ERDAS IMAGINE 9.2 (for detailed methods, see [41]).

Habitat-patch delineations from the analyzed satellite imagery were ground-truthed during the field surveys, and corrected images were used in the final analysis in GIS. The sizes (ha) of the four habitat types within each home range were calculated (in ArcGIS) during the four breeding stages to determine if the pairs were using the three habitat types differently at different stages.

A logit-distributed GLMM was also used to analyze variations in the size and proportion of habitat type within the home ranges between the four breeding stages. We specified pairs as the random effect, area/proportion of habitat type (marsh, marsh meadow, and meadow) as the dependent variable, and breeding stage as the explanatory variable.

All statistical tests were run using R version 3.20 [42]. All tests were two-tailed (α = 0.05) as no directionality was expected.

## 3. Results

Twenty-three pairs of cranes were tracked at the preincubation stage, but 7 pairs had not laid eggs, and the young had died after hatching in 3 pairs, so only 13 pairs were tracked through all four breeding stages. This yielded 268 2-h tracking sessions and 6432 GPS locations in total (mean ± SE per pair, 280 ± 129, see Appendix A).

### 3.1. Home Range

The overall mean area of the 95% FKP home-range size was 143.38 ± 34.46 ha, and the mean area of the 50% FKP was 22.89 ± 5.77 ha. The size of the 50% FKP home-range size had no significant difference between the four breeding stages (GLMM: *F*_3,63_ = 1.606, *p* = 0.197, Figure 3a). However, there were significant differences at *p* < 0.05 of the 95% FKP home-range size between the four breeding stages (GLMM: *F*_3,63_ = 4.246, *p* = 0.009, Figure 3a). Results of the pairwise comparison in the post hoc test (least significant difference (LSD)) indicated that the postfledging stage was significantly lower than the preincubation/incubation stage was (*p* < 0.001; Figure 3a).

For the different probability densities (i.e., 50% and 95% FKP), the degrees of neighboring overlap at the four breeding stages of home-range size were all lower than 10% (Table 1 and Figure 4), while the level of site fidelity varied among the four breeding stages for the different probability densities (i.e., 50% and 95% FKP); although the level of site fidelity of the 50% FKP home-range size was low (range: 10–25%), the site-fidelity level of the 95% FKP home-range size appeared relatively high (range: 35–62%), as detailed in Table 2.

### 3.2. Habitat Use

Habitat patch size within the home range of cranes did not differ in any breeding stages when using 50% FKP, but there were some differences when using 95% FKP. Cranes used smaller-sized marsh patches during the fully fledged stage than other stages. More specifically, the pairwise-comparison results in the post hoc test (LSD) indicated that marsh areas in the preincubation stage were significantly larger than those in the postfledging and fully fledged stages (*p* < 0.001, Figure 3b), and marsh areas in the incubation stage were significantly larger than those in the fully fledged stage (*p* < 0.001, Figure 3b). Cranes used smaller-sized meadow patches during the postfledged and fully fledged stage than other stages. More specifically, pairwise-comparison results in the post hoc test (LSD) indicated that meadow areas at the postfledging stage were significantly lower than those at the preincubation and incubation stages (*p* < 0.001, Figure 3b).

When comparing proportions of marsh areas within the home range, there was significant difference in the proportion of the 50% FKP marsh area among the four breeding stages (GLMM: *F*_3__,63_ = 3.273, *p* = 0.027). More specifically, pairwise-comparison results in the post hoc test (LSD) indicated that the proportion of marsh areas in the postfledging stage was significantly larger than those in the preincubation, incubation, and fully fledged stages (*p* < 0.001, Figure 5a). Likewise, there was significant difference in the proportion of the 95% FKP marsh area among the four breeding stages (GLMM: *F*_3__,63_ = 4.889, *p* = 0.004). More specifically, pairwise-comparison results in the post hoc test (LSD) indicated that the proportion of marsh areas in the postfledging stage was significantly larger than those in the preincubation and fully fledged stages (*p* < 0.001, Figure 5b). When comparing proportions of marsh-meadow areas within the home range, there was no significant difference in the proportions of the 50% or 95% FKP among the four breeding stages (GLMM: *F*_3__,63_ = 0.498, *p* = 0.685; *F*_3__,63_ = 0.680, *p* = 0.568, respectively). When comparing the proportions of meadow areas within the home range, there was significant difference in the proportion of the 95% FKP meadow area among the four breeding stages (GLMM: *F*_3__,63_ = 3.225, *p* = 0.028). More specifically, pairwise-comparison results in the post hoc test (LSD) indicated that the proportion of meadow areas in the fully fledged stage was significantly larger than those in the preincubation and postfledging stages (*p* < 0.001, Figure 5b).

## 4. Discussion

### 4.1. Black-necked Cranes Display Strong Territoriality during Breeding Season

Variations in areas of both the home ranges and three habitats (marshes, marsh meadows, and meadows) of breeding Black-necked-Crane pairs were studied during the four consecutive breeding stages: preincubation (23 pairs), incubation (16 pairs), postfledging (15 pairs), and fully fledged (13 pairs). Results of the 50% FKP indicated that the home range of the Black-necked Cranes did not change significantly throughout the four breeding stages. However, results of the 95% FKP showed that the home range of the Black-necked Cranes decreased significantly during the postfledging stage and returned to a level similar to those of the preincubation and incubation stages at the fully fledged stage. Results suggest that Black-necked Cranes may utilize different habitat-use strategies at different breeding stages.

Results showed almost no overlap between neighboring families when using 50% FKP (Figure 4 and Table 2). Meanwhile, site fidelity (%) of the 50% FKP home-range size was lower than 50%, indicating that the core region of the four reproductive stages was dynamic. However, 95% FKP home-range sizes indicated that an overlap of neighboring families existed; even the degree of overlap was low, and site fidelity was high (>50%). These results indicate that Black-necked Cranes may have strong territoriality during the breeding season. Territorial protection by Black-necked Cranes has also been recorded in the literature and via video [17]. Since the home ranges of Black-necked Cranes were relatively stable, it is particularly important to protect these areas, so as to protect the breeding of Black-necked Cranes.

### 4.2. Black-necked Cranes May Increase Marsh Utilization during Postfledging Stage

Combined with the variability result of the 95% FKP home range and marsh habitat within the home range during the four breeding stages, the proportion of marsh habitat slightly increased, while the whole range of activity decreased during the postfledging stage. These results suggested that, during the postfledging stage, the utilization of marsh habitat by Black-necked Cranes may be increased, though the trend is not strong, probably due to the small sample size. Possible explanations for increasing marsh utilization during this stage are food and safety demand from their young cranes.

(1)Demand for food by young cranes: Precocial waterfowl require an immense amount of calcium during the first month of growth due to rapid growth rates [43]. Wetlands, even high-altitude wetlands, can offer very high concentrations of food such as crabs, gastropods, and insects [44,45]. However, it is unlikely that high-altitude meadows provide an equivalent amount of such foods. The observed difference in habitats within the home range of cranes may therefore be the result of the availability of essential foods. Observations on the Black-necked Crane in other breeding areas often reported that Black-necked Cranes forage in marshes [46], which supports this view. Consequently, differences in marsh area within the measured home range could be related to the availability of calcium-rich foods.(2)The demand for safety by young cranes: The increased utilization of marshes by Black-necked Cranes during the postfledging stage may also be due to safety concerns [47]. Water depth and vegetation height in marsh habitats significantly exceed those in marsh-meadow and meadow habitats (see Appendix A). Compared with marsh meadows and meadows, marshes may be a safer environment, which may have been the initial reason why marsh habitats were chosen for nesting [23]. During the postfledging stage, young cranes have just emerged from their shells, and their mobility is weak. In the Zoige wetland, likely natural enemies during this period are the Tibetan Fox (*Vulpes ferrilata*), the Red Fox (*Vulpes vulpes*), and the Upland Buzzard (*Buteo hemilasius*). Therefore, to protect young cranes, their parents tend to reduce the overall range of activities and choose safer areas for their activities. Other birds also display similar tendencies, such as the Black-tailed Godwit (*Limosa limosa*) in the Netherlands, where families with chicks stay mainly in herb-rich fields [9]. Therefore, food and safety may be the main driving factor in the selection of this habitat during the postfledging stage.

### 4.3. Role of Meadows and Marsh Meadows during Breeding Season

The area and proportion of meadow habitats within their home ranges were the largest among all four breeding stages. A significant decline was observed during the postfledging stage, which was consistent with the variation of the home range. However, marked recovery was found at the fully fledged stage. This indicates that Black-necked Cranes make different choices for meadow utilization during the two stages. One reason why more meadow habitats were utilized during the fully fledged stage may be food. Previous studies showed that Black-necked Cranes mainly forage in meadow habitats, which are abundant in vertebrate and invertebrate food resources. Cow dung in meadow habitats can provide an abundance of food for invertebrates [48]. Grazing may also indirectly provide shelter for invertebrates by changing the immediate environment around cow dung [49]. Therefore, grazing may indirectly provide a steady food supply for Black-necked Cranes. It is, therefore, very important to study the relationship between grazing activities and the protection of Black-necked Cranes for their conservation.

Another reason for increasing meadow-habitat utilization at the fully fledged stage might be that there are few wetland resources in the Zoige wetland, so Black-necked Cranes can only forage in meadow habitats. In other breeding areas of Black-necked Cranes, such as Bange county, Tibet, and the Yanchiwan national nature reserve, Gansu, cranes can often be observed to forage in marshes [46]. Alpine wetlands on the Qinghai–Tibetan Plateau are important breeding and staging areas for several endangered wetland-bird species [50]. Examples are Baer’s Pochard (*Aythya baeri*), the Baikal Teal (*Anas formosa*), the Marbled Teal (*Marmaronetta angustirostris*), and the Black-necked Crane. As a result of global warming, artificial drainage, and intensified grazing activities, Zoige wetlands have severely retreated [28]. Marshes have degenerated into marsh meadows or meadows [51]. The significant reductions in the wetlands on the Qinghai–Tibetan Plateau threaten ecosystem function [52], thus further endangering waterfowl breeding. The number of Black-necked Crane populations has increased in recent years [53]. This suggests that intraspecific competition for marsh resources may be the most crucial issue at present. Therefore, the coordinated and sustainable development of human communities (and associated activities), and the conservation of the Qinghai–Tibetan Plateau alpine wetlands and water birds are imperative [54]. The protection of marshes may be an indispensable part of the protection of the Black-necked Crane.

Relevant studies showed that birds move long distances to find suitable foraging sites during late puberty stages [6]. Both the size and proportion of marsh meadows did not significantly change during the four consecutive breeding stages. Marsh-meadow habitats may act as a corridor for Black-necked Crane families to move from marsh to meadow. Studies showed that conserving this corridor is very important for birds [10,13]. Consequently, the protection of marsh-meadow habitats for cranes in the future is an important issue.

Relevant studies showed that birds move long distances to find suitable foraging sites during late puberty stages [6]. Both the size and proportion of marsh meadows did not significantly change during the four consecutive breeding stages. Marsh-meadow habitats may act as a corridor for Black-necked-Crane families to move from marsh to meadow. Studies showed that conserving this corridor is very important for birds [10,13]. Consequently, the protection of marsh-meadow habitats for cranes in the future is an important issue.

### 4.4. Influence of Grazing Activities during Breeding Stages

Although agricultural activities are generally recognized as an important factor for the loss of bird diversity [55], the relationship between agricultural (especially grazing) activities and bird protection is still a hot issue [56]. In meadow habitats, mixed grazing at low intensity provides a better foraging environment for birds by increasing habitat heterogeneity, which modifies invertebrate accessibility; thus, bird populations are ultimately increased [57]. However, in marsh habitats, grazing may cause the eggs to be trampled or picked up by yaks and herdsmen [46].

According to observations from 2012 to 2014, the activities of young cranes often occur in marshes, and the mortality rate of young cranes is high (53%) during postfledging [58]; therefore, limiting grazing in the marshes during this stage may effectively improve the survival rate of young cranes. Considering that marshes occupied a substantial area during the incubation period, grazing in the marshes should also be monitored during this stage. Thus, we suggest that grazing activities in the marshes from late May to early September be carefully managed for the successful breeding of Black-necked Cranes.

## 5. Conclusions

We investigated the home range and habitat use of the Black-necked Crane during different breeding stages at the Zoige wetland. Throughout the whole breeding period, home-range size has a trend of first decreasing and then increasing. The degree of overlap between the home ranges of neighboring crane families was small; however, site fidelity was high. Consequently, Black-necked Cranes may display strong territoriality during the breeding season. Black-necked Cranes mainly utilize meadows in the whole breeding season, followed by marsh meadows and marshes. Although marshes are not the main utilization habitat for the breeding Black-necked Cranes, cranes slightly increase their utilization proportions of marshes during the postfledging stage. Since this stage is crucial for young cranes’ survival, further studies including more crane families should be conducted to evaluate the importance of marshes to the conservation of Black-necked Cranes.

## Figures and Tables

**Figure 1 animals-10-01975-f001:**
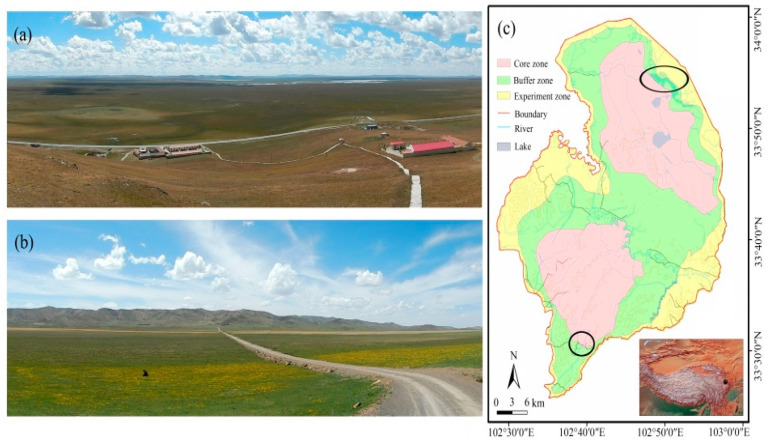
Study area in Zoige wetland on the Qinghai–Tibetan plateau. (**a**) Flower marsh (photo taken on 14 September 2015); (**b**) Naleqiao marsh (photo taken on 20 May 2016); (**c**) map of Zoige wetland nature reserve (upper black ellipse, Flower marsh; lower one, Naleqiao marsh).

**Figure 2 animals-10-01975-f002:**
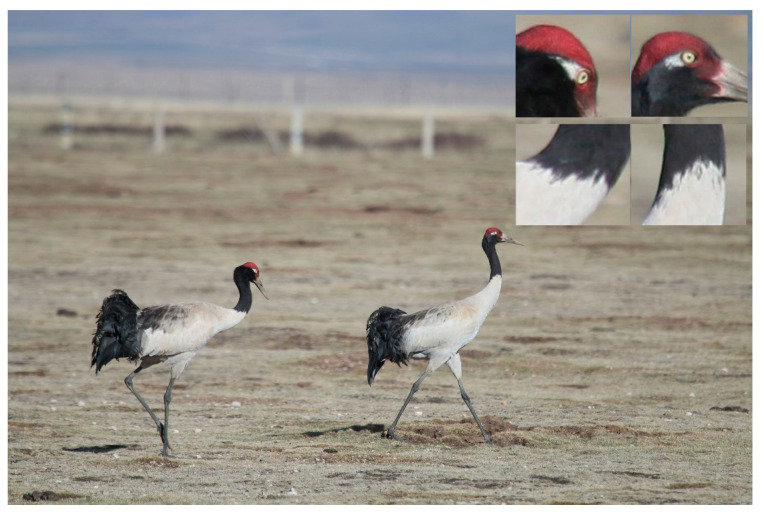
Irregular edge between black and white on necks and behind eyes of Black-necked Cranes (taken on 14 September 2015).

**Figure 3 animals-10-01975-f003:**
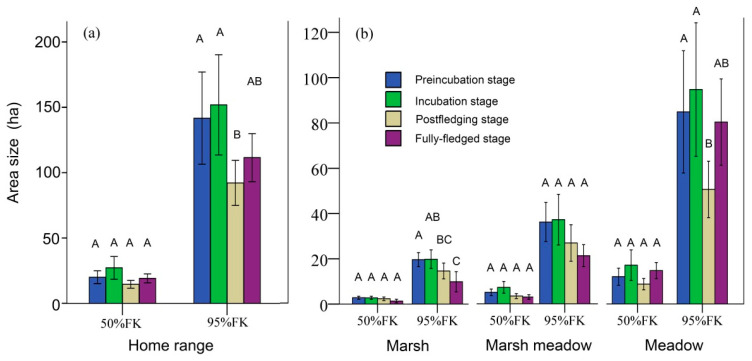
Differences of (**a**) home-range and (**b**) habitat-patch sizes at four breeding stages (standard error (±1 SE) represented by vertical bars). Letters above bars (A, B and C), pairwise significant differences; bars with different letters, significant at *p* < 0.05; bars with similar letters, not significantly different from one another (*p* > 0.05).

**Figure 4 animals-10-01975-f004:**
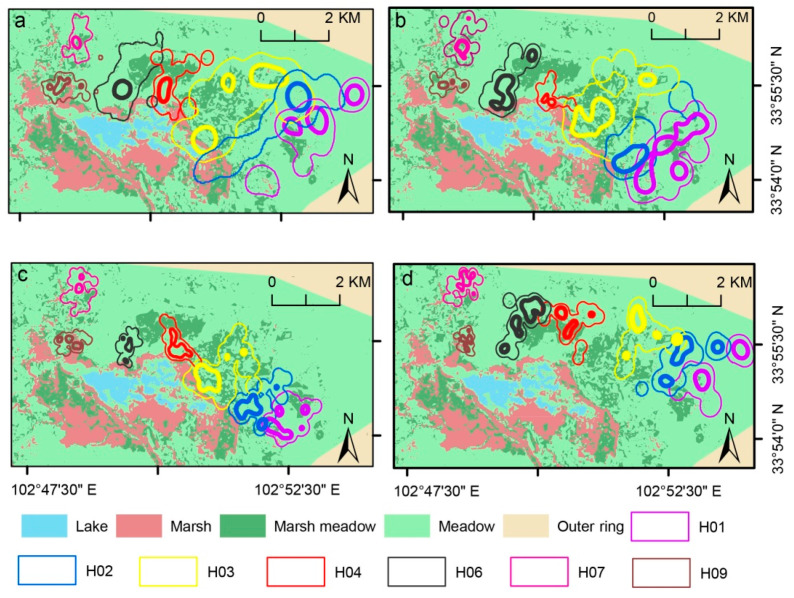
Seven of 13 Black-necked Crane pairs’ home-range areas (e.g., H01) during four stages ((**a**), preincubation; (**b**), incubation; (**c**), postfledging; and (**d**), fully fledged) at (**a**) Flower marsh in study area. Differently colored lines, home ranges of different breeding pairs; thick lines of same color, 50% FKP; corresponding thin lines, 95% FKP.

**Figure 5 animals-10-01975-f005:**
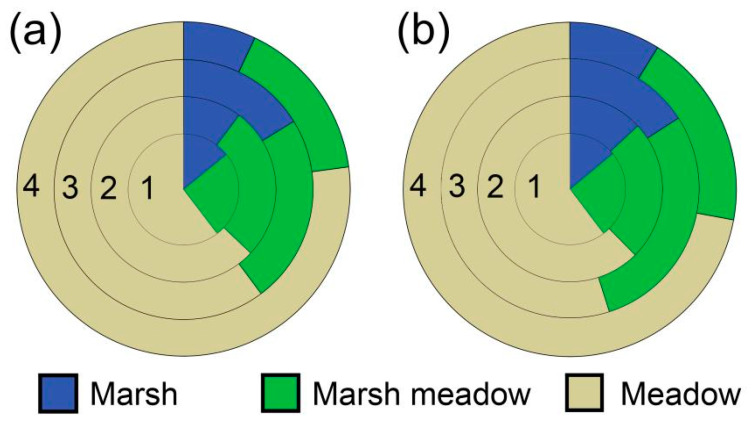
Area proportions of three habitats (marshes, marsh meadows, and meadows) within home ranges of Black-necked Cranes during four breeding stages: 1, preincubation; 2, incubation; 3, postfledging; 4, fully fledged; (**a**) 50% FKP; (**b**) 95% FKP.

**Table 1 animals-10-01975-t001:** Home-range overlap areas of neighboring Black-necked Cranes. FKP, fixed-kernel polygons; PI, preincubation; IN, incubation; PF, postfledging; FF, fully fledged; N0, number of neighboring families (nonrepetitive counting); and N1 and N2 numbers of neighboring families with overlapping areas.

		50% FKP				95% FKP		
	N0	N1	Overlap Area (ha, Mean ± SE)		Overlap Degree	N2	Overlap Area (ha, Mean ± SE)	Overlap Degree
PI	26	0		0	0	22	17.83 ± 7.9	0.08 ± 0.02
IN	11	0		0	0	9	19.03 ± 9.01	0.09 ± 0.02
PF	10	1		0.12	0	7	7.49 ± 2.93	0.08 ± 0.02
FF	9	1		0.65	0.04	7	9.87 ± 3.82	0.06 ± 0.02

**Table 2 animals-10-01975-t002:** Site fidelity of Black-necked Crane among four breeding stages (mean ± SE). FKP, fixed-kernel polygons; PI, preincubation; IN, incubation; PF, postfledging; and FF, fully fledged.

	50% (*n* = 13)		95% (*n* = 13)	
	Overlap Area (ha, Mean ± SE)	Site-Fidelity Level	Overlap Area (ha, Mean ± SE)	Site-Fidelity Level
PI: IN	6.06 ± 2.43	0.22 ± 0.06	116.06 ± 36.24	0.62 ± 0.03
PI: PF	3.01 ± 0.93	0.2 ± 0.06	64.68 ± 16.98	0.5 ± 0.05
PI: FF	5.17 ± 2.48	0.19 ± 0.06	79.95 ± 19.45	0.53 ± 0.06
IN: PF	6.12 ± 2.54	0.25 ± 0.06	71.78 ± 19.02	0.56 ± 0.03
IN: FF	2.42 ± 1.23	0.1 ± 0.05	56.44 ± 12.99	0.45 ± 0.06
PF: FF	1.1 ± 0.57	0.1 ± 0.05	27.98 ± 4.11	0.35 ± 0.06

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
