# Peer review of "Home Range and Habitat Use of Breeding Black-necked Cranes"

_animals, 2020, doi:10.3390/ani10111975_

Round 1

Reviewer 1 Report

Thank you for your responses to my prior edits and I believe the paper is much improved.  Minor comments at this point:

Title: I think this is an improvement to the title.  I think the title could be even stronger if you use increase rather than increased (i.e., black-necked cranes increase marsh utilization during the post-fledging), but either way is fine.  

Line 13: My original question was "which plateau?" and was just referring to the Qinghai-Tibetan plateau. I was meaning to suggest that the name of the plateau be added here since it is the first sentence the reader sees of the entire paper.  Your response indicates, if I am understanding correctly, that this is the only crane that breeds on plateau habitat at all, which is a different issue.  If that’s correct, then I think the text can remain as you have it now. 

Line 45:  please add an s to the end of area.

Figure 1: The Figure is much improved.  I should have asked this the first time, but what is meant by “experiment zone” in this figure?

Line 207: should now say “times during each stage” rather than “times each stages”.

Line 214: should say significant differences rather than difference.

Line 304: should say decreased rather than decreasing.

Thank you for the explanation and image of your camouflage field clothes, I love it.  This does sound like such fun fieldwork to do.

Reviewer 2 Report

Evidently the English language editing services provided by Wiley are not very thorough, so I have taken the time to suggest changes (listed below) to the Title, Simple Summary, Abstract, and Introduction so that your paper will get off to a good start.

Please make the following changes:

Title: Please change “increased” to “increase”

Line 13: The Black-necked Crane is the only crane that breeds on the Qinghai-Tibetan Plateau, and is currently classified as a vulnerable species.

Line 45: , and one of the major Chinese pastoral areas.

Line 48: has been a focus of wildlife conservation…

Reviewer 3 Report

The authors report to “have substantially revised the manuscript following your suggestions”. I’d question “substantially” as quite a lot of points have been ignored.

In my review of the original submission, I raised three major concerns.

  1. key conclusions and management recommendation is, at best, only very weakly supported by the data.

This remains the most substantial stumbling block. The authors seem intent on making a point for management which might be quite correct and very pertinent to conservation but for which they don’t provide substantive support.

  1. argument about territoriality may be circular.

This is much improved and is now close to adequate.

  1. poor presentation.

In some respects, the standard of presentation is much improved. The Results are clearer, the statement of aims is better, and the sub-section headers are now intelligible. The general flow of the text is considerably improved though it couldn’t be regarded as excellent. There are two remaining issues:

- the authors have reduced the number of references cited from 117 to 97. This still seems grossly excessive to me, but that’s ultimately your call as I know journals vary in their approach to this issue.

- on the question of explaining the three habitats, the authors response is unsatisfactory, as explained in my notes for them.

There are a number of minor points that remain to be satisfactorily addressed.

In conclusion, whilst most remaining issues could be resolved with further revision, point 1 above is fundamental and I think calls for rejection of this manuscript in its current form. You could invite resubmission along the lines I’ve suggested to the authors.

Round 2

Reviewer 1 Report

I have no additional comments.

Author Response

Thank you for giving support to accept your paper.

Reviewer 2 Report

Overall the authors have made substantial improvements. However, the following is not correct and I pointed this out in my previous review and it was not addressed.  You cannot just say "on plateau" because 1) it is not grammatically correct and 2) you have to be specific about what plateau you are referring to (i.e., the Qinghai-Tibetan Plateau).

"The Black‐necked Crane is the only crane that breeds on plateau,"

Author Response

Thank you for your suggestion and for giving support to accept our manuscript. We have revised by the suggestion. (Line 13)

Reviewer 3 Report

I appreciate that you’ve altered your title and aims, reduced the number of references, clarified the statistical analysis and attended to other minor items. However, your refocussing of the manuscript seems half-hearted and isn’t convincing. For instance, your Abstract and Discussion don’t primarily reflect a story about habitat use in total; you continue to pick out your poorly supported story about marsh habitat as a strong focus.

The presentation remains quite poor in places. There are numerous instances of awkward expression – this needs a really thorough edit by a fluent scientific writer. The following are examples only and do not preclude the need for a really thorough edit:

* your reference to “Data structure subjects”. I don’t know what you mean. These phrases/sentences could simply be deleted.

* the wording of the first paragraph of the Results is still confusing. The problem probably harks back to the Methods, where the relevant quantification of effort is dispersed though sections 2.2 and 2.3, and in the beginning of 2.3 we have “four times” but only later in the paragraph is it explained that a “time” actually means a 2-hour session in which 24 locations were obtained, not “one time”. If you re-write the methods section using clear and consistent terminology and with the relevant information about quantification aggregated, then I think the first paragraph of the Results becomes something like (with the appropriate number inserted at “xx”:

“Twenty-three pairs of cranes were tracked at the pre-incubation stage, but seven pairs didn’t lay and the young died after hatching in three pairs, so only 13 pairs were tracked through all four breeding stages. This yielded xx 2-hour tracking sessions, and 6,432 GPS locations (280 ± 129 per pair).”

* The Conclusion is very poorly worded.

* It took me ages to work out the meaning of Appendix 1, apparently because the methods are offset by two lines from the metrics they apply to.

Author Response

This manuscript is a resubmission of an earlier submission. The following is a list of the peer review reports and author responses from that submission.

Round 1

Reviewer 1 Report

My comments are primarily minor editorial changes that would improve the clarity of the paper.  I really enjoy the simplicity of this study and find the paper to be generally well written and well referenced.  As mentioned in a couple of my comments, I'd like to know more about the study area in general, as a reader from a different continent unfamiliar with this ecosystem but one with a strong appreciation for cranes.  The fieldwork, as described, seems like it would be highly enjoyable.

Manuscript comments

  • Line 13, remove s on the end of cranes.
  • Line 13, which plateau?
  • Line 18, comma needed after “that”
  • Line 19, do you mean increased rather than improved?
  • Line 25, important grazing area for which livestock? Cattle?
  • Line 36, again I am not sure of the use of the word improved in this sentence. Improved implies a value judgement about their use of marsh habitat.  I would just say increased.
  • Lines 48-53, can you elaborate (for someone from North America) what species are grazed on the plateau and what the specific effect of their grazing activities does to the habitat for the birds?
  • Line 63, change statistics to characteristics? Biology?
  • Line 71, remove comma after network.
  • Lines 73-74, these two sentences seem to conflict. The first one indicates that habitat loss and destruction are the main threats and the second one then indicates that the most urgent need is research.  Wouldn’t the most urgent need be habitat protection if habitat loss is the largest problem?  Or can you elaborate on what specifically is needed from research in order to enhance/improve habitat protection?  You do get to this in the subsequent paragraph but maybe the two paragraphs need to be tied together better.
  • Line 76, probably a dumb question but again, for someone not familiar with the region, I assume the Zoige wetland is on the Qinghai-Tibetan plateau? This is explained eventually in the study area but just a quick mention that it is on the plateau would be helpful.
  • Line 82, remove the space before the period.
  • Line 85, why changes of nesting area specifically? Are they changing from one type to another during the season?  Do you just mean the characteristics of the nesting habitat?
  • Line 93, change rather to often.
  • Line 97, add s to affect.
  • Line 98-100, change the second understand in this sentence to elucidate or clarify.
  • Line 108, husbandry of what species?
  • Line 109, change to “To study the needs of the crane during different reproductive stages…” The need is on behalf of the crane, not the reproductive stage itself.
  • Line 113, change existed to exist so that you have the same tense in both hypotheses.
  • Lines 119-120, rephrase to “hotspot inhabited by a number of endemic and endangered…” Inhabited at the end of the sentence as currently written does not make sense.
  • Line 129, insert of after habitats.
  • Lines 135-135, left parenthesis missing on a), there is no reference to or explanation of part (c) of this figure, and the image in the bottom right of part (c) is too small to be able to tell what it is showing.
  • Lines 147-154, there is no reference to Figure 2 anywhere in the body of the text of this paragraph although Figure 2 appears below it. Also, is the reader to understand from this paragraph that individual cranes have feather patterns distinct enough that they can be identified throughout the season this way?  Is this a common methodology for identifying individuals of this species? 
  • Lines 156-158, a few questions. What is meant by installing a satellite positioning system?  Does that mean capturing the birds and outfitting them with some kind of GPS device or does it mean something else?  This is a little bit confusing because 13 pairs actually seems like a small sample size, but I am not sure exactly what a satellite positioning system refers to and why it is not conducive to the protection of local populations.  Can you elaborate on this?  In any case, the method you did use of following and observing birds sounds like so much fun!
  • Line 167, should say we recorded our own position.
  • General question: how tall is the vegetation in these study locations and how visible do you believe you were to the cranes, if at all. Is there any possibility that your presence affected their behavior and space use?
  • Line 191, either omit the word more from this sentence or identify specifically what you mean by more flexible approach – more than what?
  • Line 192, what is an identity linked GLMM; what specifically does the identity link part mean or what does it do that is different from a standard GLMM?
  • Line 229, should say stages instead of stage.
  • Line 244, should say are significantly different at P < 0.05.
  • General comment: Table 1 and Figure 3 are representing the same information, I do not think that both are needed, I think you could use one or the other rather than both. In fact I would suggest a careful scrutiny of all of the tables and figures to make sure you are not representing redundant information in them; some may be possible to omit.
  • Lines 278-279, it is generally a bad approach to make a statement like “Table 1 shows”… etc. Better to make a statement about the finding you want to share here and just reference the table parenthetically.  For example, “home range size was largest in the incubation stage (Table 1).”
  • Lines 286-287, add this to previous paragraph, should not be its own one-sentence paragraph.
  • Lines 311-312, same comment, avoid using one-sentence paragraphs.
  • Line 321, habitat should be plural.
  • Lines 330-331, I would recommend omitting the sentence referencing a follow-up paper.
  • Line 336, change indicated to indicate.
  • Line 345, I would change this to say use of marsh habitat was increased or was higher rather than strengthened. Strengthened is awkward in this context.   
  • Line 359, change to “The increased utilization of marshes” and omit “was enhanced”
  • Line 372, I really like how you have used active voice in the two preceding subheadings (e.g., cranes display strong territoriality (4.1), marsh utilization is strengthened (4.2)). Is there a way to also make this subheading active voice, i.e., “marsh meadow and meadow are…”
  • Line 394, another reason for what specifically? Can you rephrase this sentence since it is the first one in the paragraph?
  • Line 399, should read wetlands have retreated and marshes have degenerated.
  • Line 400, reproduction seems like an odd word choice in this sentence. Maintenance of water flow?
  • Lines 409-411, can you elaborate on your logic between these two sentences? Why does an increased population in recent years specifically indicate that intraspecific competition for marsh resources may be the most critical issue?
  • Line 430, populations should be plural.
  • Line 433, should say grazing enhances habitat heterogeneity.
  • Line 435, when determining bird population what? Abundance?  Trends?
  • Line 436, what method specifically are you referring to? This sentence and the one before it do not follow one another very clearly.
  • Lines 437-439, add an s to provide and to population.
  • Line 440, omit the word out.
  • Line 441, either omit the word put, or rephrase to put in place and omit issued.
  • Line 442, rephrase to Based on the results of this study,…
  • Line 447, you mention “if conditions permit.” Can you elaborate on this?  This could be done here or earlier in the paper but as a reader from a different continent and culture, I would really love to know more about the land ownership/management, which livestock are grazed on these lands and for what specific purposes (we don’t find out until the discussion that the grazing referred to throughout relates to cows, and also possibly yaks), and what the possibilities are for prohibiting grazing during certain times of the year?  How feasible/challenging is that recommendation to implement in your system?  Is there high interest in the birds and their conservation among the local communities? 
  • Line 449, should say prohibiting grazing during this stage may also improve success rate of hatching, correct?
  • General comment: Appendix 2, while interesting, is not really necessary to the paper.

Author Response

Thank you for the opportunity to revise our manuscript. We have substantially revised the manuscript following your suggestions.

Reviewer 2 Report

This manuscript, describing aspects of reproduction in Black-necked Cranes, is scientifically sound, but poorly written. The introduction especially, could be significantly condensed as at least half (if not more) is absolutely redundant. That is, the same statements of fact are repeated several times (lines 41-103). Additionally, lines 109- 144 belong in the methods section, which would result in lines 141-143 being redundant. This manuscript also requires further English language editing before publication. Finally, a specific statement regarding what animals are grazed in the study area is needed (cows, sheep, ?).

Simple Summary: Please explicitly state what plateau you are referring to on line 13. The summary must be able to stand alone as an ample description of your work.

Title (and throughout the entire manuscript): The use of the words "strengthen"  "enhance", "improved", etc... are not appropriate in reference to the "increased marsh habitat utilization". Perhaps the authors were trying to avoid sounding redundant with their word use, but "increased" is the appropriate term in this instance. 

Again, this work is scientifically sound and an interesting contribution worthy of publication. However, before I can recommend publication, editing by a Native English speaker with experience in science writing is required (preferable an ornithologist).

Author Response

(The authors gave the same response as above.)

Reviewer 3 Report

This is an interesting and potentially valuable contribution to crane ecology. I have three major concerns.

The first is that you’ve presented the paper and your key management recommendation around a pattern that is weak. The cranes did not use more marsh at the post-fledging stage (Fig. 3), only proportionately more, and the latter trend was weak (overall P = 0.03) and marsh was never more than a minor component of their habitat (Fig. 5).

My second concern is that you argue for strong territoriality but it seems to me that there’s a risk of circularity in the argument. How did you identify your pairs particularly once the young had left the nest? Was it by location? But perhaps you can address this by telling us how you identified pairs independent of location.

The third is that your presentation is quite difficult to follow. To make it much easier to read, I recommend a thorough re-draft including the following:

* in the Results, move as many statistics as possible to tables or figures, concentrating in text of the key messages arising;

* use much fewer references. I note 117 listed references, a quite excessive number. I suggest asking yourself, for each citation, does it actually strengthen the manuscript? As a rule-of-thumb I’d think 40 references would be a good number to aim for with 60 a maximum for this type of manuscript;

* match your hypotheses to the issues you address (see comment below);

* use simple sub-section headers in place of the long and convoluted 3.1., 3.2. and 3.3.**; and

* a thorough edit of your manuscript at the level of concepts, sentences and paragraphs, avoiding unhelpful duplications and opaque concepts – some specific examples and suggestions below.

** for example, 3.1. could be simply “Home range” and 3.2. and 3.3. combined as “Use of habitat”.

Given the centrality of the three habitats to your study, your readers need considerably more detail about them and how they were determined than you’ve provided (l132-3).

Examples of writing style issues, first paragraph of Introduction only

l42-6: the second sentence more or less just repeats the first one

l46: the phrase “According to relevant research” is redundant and should be deleted

l48-50: this sentence is awkward. How about “Conservation biologists are often required to resolve conflicts between agricultural intensification and the needs of wildlife (~).”

l51: the phrase “accounts for half of China's major pastoral area” in opaque.

l51-3: poorly worded. I suggest replacing this sentence with “In recent years, an increase in grazing activity, along with reduced area of wetland, has reduced habitat for resident birds (~).”

l54-6: poorly worded. I suggesting replacing this sentence with “Finding a balance between grazing and the protection of rare species has been a focus for wildlife research on the Qinghai-Tibetan Plateau (~).”

Minor comments

Title and elsewhere: “strengthen marsh utilization”. This phrase is unclear. I guess you mean they made more use of marsh at that stage.

Abstract: the presentation of statistics here is unhelpful, and in the form you’ve presented it is misleading because select pairwise comparisons doesn’t reflect your overall data analysis. For example, it makes the proportional use of marsh look like a much stronger trend than it is.

l111-4, hypotheses.

- these sound like null hypotheses (H0), not hypotheses (H1). Indeed, your data analysis is structured so that these are null hypothesis.

- personally, I’d rather you structured these as questions, e.g. does home range size change over the course of breeding?

- the sentence doesn’t make sense; I think you mean “were addressed” or similar

- further, the two hypotheses don’t match the three general propositions (the first with sub-propositions) presented in your Results section.

“post-fledging” (l144-5). Fledging is when the young leave the nest, not when they hatch. Do they leave the nest immediately upon hatching?

Fig. 1:

- what are the black ellipses? I can only guess. Need to say what they are.

- you illustrate “core”, “buffer” and “experiment” zones, concepts that don’t appear in your text even in a management context. What is the point of these?

Fig. 2: what is the point of this? I don’t mind illustration, but it needs to address a point raised in your text.

l156-7: the arguments are not elaborated and opaque – it isn’t clear why they’re “unrealistic” and “not conducive”. You could explain these, but I suggest simply deleting the arguments.

l191-5:

- is the first sentence really necessary or helpful?

- I can’t see what is nested in what. Don’t you have a repeated measures design with pair as the repeated measure (random effect)? But I’m willing to be corrected about this.

lines 239-40 etc., sample size:

- I don’t understand lines 239-40. If there are 96 “fixed GPS locations” then what is N = 6,432?

- I further eventually found the key information about sample size – the number of 2-hour monitoring periods per pair per breeding stage – in lines 158-9, but even here it is not clearly linked to “2-hour observation session” later in the paragraph.

- sample size considerations (l239-42) surely belong either in your Methods section or above the first sub-head in the Results as they apply to all propositions, not just the propositions in section 3.1, with all relevant information coherently collated and summarised together.

Table 1 appears to be exactly the data for Fig. 3. This is avoidable duplication. I urge you to move Table 1 to supplementary data.

Table 2 and associated text. I’m not clear what has happened with the division of this table across pages. What is “%”? Aren’t the overlap figures proportions? Either way, need to be clear about this and consistent between the table and text so as to make it easy for your readers.

- ditto Table 3: here’s you’ve overtly confounded proportions and percentages: your site fidelity data are presented as proportions but the header states that they’re percentages.

Fig. 4, line 274: “Different colours” should be “Different colour lines”

I feel that Appendix 1 would benefit from a title explaining its contents.
